# On-Field Biomechanical Assessment of High and Low Dive in Competitive 16-Year-Old Goalkeepers through Wearable Sensors and Principal Component Analysis

**DOI:** 10.3390/s22197519

**Published:** 2022-10-04

**Authors:** Stefano Di Paolo, Francesco Santillozzi, Raffaele Zinno, Giuseppe Barone, Laura Bragonzoni

**Affiliations:** Department for Life Quality Studies, University of Bologna, Via di Barbiano 1/10, 40136 Bologna, Italy

**Keywords:** goalkeeper, football, dive biomechanics, wearable sensors, principal component analysis, ecological dynamics

## Abstract

Diving saves are the main duty of football goalkeepers. Few biomechanical investigations of dive techniques have been conducted, none in a sport-specific environment. The present study investigated the characteristics of goalkeepers’ dive in preferred (PS) and non-preferred (nPS) side through an innovative wearables-plus-principal-component analysis (PCA) approach. Nineteen competitive academy goalkeepers (16.5 ± 3.0 years) performed a series of high and low dives on their PS and nPS. Dives were performed in a regular football goal on the pitch. Full-body kinematics were collected through 17 wearable inertial sensors (MTw Awinda, Xsens). PCA was conducted to reduce data dimensionality (input matrix 310,878 datapoints). PCA scores were extracted for each kinematic variable and compared between PS and nPS if their explained variability was >5%. In high dive, participants exhibited greater hip internal rotation and less trunk lateral tilt (*p* < 0.047, ES > 0.39) in PS than nPS. In low dives, players exhibited greater ipsilateral hip abduction dominance and lower trunk rotation (*p* < 0.037, ES > 0.40) in PS than nPS. When diving on their nPS, goalkeepers adopted sub-optimal patterns with less trunk coordination and limited explosiveness. An ecological testing through wearables and PCA might help coaches to inspect relevant diving characteristics and improve training effectiveness.

## 1. Introduction

The diving save is the most frequent and critical task for football (soccer) goalkeepers [1,2]. Goalkeepers’ dives are explosive defensive jumps aimed at reaching the ball and preventing the opposite team from scoring [3]. For keepers, saving goals is as fundamental as scoring is for strikers [2]. For these reasons, measuring the movement quality and effectiveness of dives allows to establish goalkeepers’ performance and determine their value [2,3], especially in football academies. 

Currently, the goalkeeper’s movement assessment and training are based on qualitative visual analysis and rely on the experience of the coach [4]. However, given the complexity and the multifactorial nature of the task, quantitative assessments identifying technique deficiencies should be provided to support coaches to improve training. In this context, the comparison of kinematical differences between the preferred (PS) and non-preferred (nPS) diving side might provide information on how to equalize and improve diving performance [2]. 

Studies regarding goalkeepers’ diving are limited to the investigation of penalty kick strategies [3,5,6] and the quantification of anticipatory times [2,7]. Indeed, the literature on goalkeepers’ kinematics is scarce and lacks sample power (most of the studies have a sample size between 6 and 11 players), ecological dynamics perspective (all in-lab settings), and involvement of academy players (all >18-years-old players involved) [1,2,3,8]. Moreover, no previous study accounted for full-body kinematics or tried to extract “relevant” features from the multitude of variables involved. Consequently, limited practical implications have been provided to improve diving movement quality and performance [1,8].

Principal component analysis (PCA) has been increasingly used to deal with data multidimensionality in movement analysis. Such a technique allows the description of characteristics of multiple kinematic waveform through a limited number of uncorrelated informative features (principal components) that can express relevant similarities or dissimilarities when speeding up the interpretation process [9,10,11,12]. The assessment of diving characteristics through an ecological dynamics approach coupled with data dimensionality reduction might help to describe and quantify the biomechanical features associated with diving-side preferences. Previous studies investigated on-field biomechanics of outfield football players during sport-specific activities, such as agility tasks and side-games, and put movement characteristics (also extracted through PCA dimensionality reduction) in relation to players’ coordinative abilities, training load, fatigue, and lower-limb injury risk [13,14,15,16].

Therefore, the purpose of the present study was to investigate the underlying relevant motion characteristics of academy goalkeepers’ dives in a sport-specific scenario in relation to their preferred and non-preferred side by means of wearable technology and PCA. It was hypothesized that (I) lower limb and trunk kinematical differences would exist in frontal and transverse plane between the preferred and non-preferred side; (II) differences would exist in terms of both ipsilateral and contralateral limbs [2].

## 2. Materials and Methods

### 2.1. Participants

Nineteen competitive academy goalkeepers (16.5 ± 3.0 years, Tegner Level 9) were enrolled (Table 1). At the time of data collection, all players were under contract by a football team, actively playing at least one match per week (plus two-three trainings), had at least five years of experience as a goalkeeper, and had no history of previous severe musculoskeletal injury (intended as >28 days off). The Bioethical Committee of the University of Bologna (ID: 25861 10 February 2020) approved the present study. Each player’s parent/tutor signed informed consent before enrolment. An experienced goalkeeper’s coach (M.V.) supervised the data acquisition process.

Although the majority of previous studies on goalkeepers’ diving biomechanics were conducted on a sample size between 6 and 11 participants [1,2,3,8], one recent study estimated that a sample size of 16 participants would be required to obtain a power of 0.8 with an effect size of 0.25 and a alpha level of 0.05 [6].

### 2.2. Data Collection

The data collection was performed off season during summer break. All tests were conducted in daylight. Every participant performed a series of high and low dives on their PS and nPS. The dives were performed in a regular football goal (7.32 m × 2.44 m) on a natural grass football pitch. In each trial, the ball was positioned in one of the four corners of the goal: for the low dives, the ball was positioned 1 m in front of the goal line and 1.5 m away from the goalpost; for the high dives, the ball was tied to the crossbar through a 1.5-m rope at 1.5 m away from the goalpost (Figure 1). The players’ starting position was moved back by 1 m to allow a more natural movement (not “purely” lateral) and avoid the risk of crashing the goalpost when falling. As an example, for a dive save on the right side (goalkeepers’ right side), the players’ starting position was 1 m backwards the goal line and 2 m away from the left goalpost (Figure 1). Mattresses were placed below the fall areas of the players to avoid injuries to the goalkeepers and damages to the wearable sensor units. Overall, three valid repetitions per side, 12 trials in total, were collected (Figure 2). An experienced goalkeepers’ coach (M.V.) instructed the players and checked the validity of each trial. The same coach whistled to give the start to each trial.

Full-body kinematics (ankle, knee, hip, pelvis, trunk, shoulder, elbow, wrist joints and head) of the diving saves was collected through a set of 17 wearable inertial sensors (MTw Awinda, Xsens Technologies, Enschede, The Netherlands). The sensors’ placement was performed by a single experienced operator (S.D.P.) according to the manufacturer guidelines. In brief, sensors were placed bilaterally on feet (middle bridge), shanks (shin bone), tights (lateral side), shoulders (scapulae), arms (lateral above elbow), forearms (lateral below elbow), and hands (backside); the pelvis sensor was placed on the sacrum, the trunk sensor was placed on the sternum, and the head sensor was placed through a headband (Figure 3). Data were collected at a sampling frequency of 60 Hz. A static (upright standing) and dynamic (walking) system calibration was applied before the first trial and repeated if the position of a sensor accidentally changed during a trial. The wearable system used had been previously proved to be accurate and reliable in the assessment of high dynamics movements [17,18]. A frontal view video capture through a smartphone (iPhone 12, Apple Inc., Los Altos, CA, USA) was collected for each trial and used to help identifying the diving phases (see Section 2.3).

A set of strength tests was also conducted: maximal double-leg hop for distance, maximal single-leg hop for distance (both preferred and non-preferred sides), maximal 5 m frontal sprint, maximal 5 m lateral sprint (both preferred and non-preferred sides). The distance reached in the hops and the time elapsed for the sprints were collected. Each player performed three repetitions of each test, and the best performance was kept.

### 2.3. Data Analysis

Data analysis was conducted in a custom Matlab script (v2022a, The MathWorks, Natick, MA, USA). Kinematic data (°) were normalized between two specific frames of the diving save: the starting frame (0% of the movement) was chosen as the initial contact of the contralateral foot (CF_IC_, contralateral to the diving side) and the final frame (100% of the movement) was chosen as the frame the player touched the ground (specifically, the mattress) or remained still after the save, in high and low dive respectively. For each trial, a further four frames were identified: contralateral foot toe-off (CF_TO_), ipsilateral foot initial contact (IF_IC_), ipsilateral foot toe-off (IF_TO_), ball contact (Ball) [2,7,8]. The frames were detected through the “foot contact point detection” variable and trough visual inspection of the avatar reconstruction provided in the manufacturer software environment. The smartphone video captures were used to support the detection of the foot contact phases and dive peculiarities (undesired initial steps, hand-ball contact, etc.). The identification was carried out in the Xsens software (v2021.0.1) environment by a single operator (F.S.). Through these frames, it was possible to identify six meaningful movement sub-phases according to the literature: Initiation phase (from CF_IC_ to IF_IC_ [2,8]); Take-off phase (from CF_TO_ to IF_TO_ [2,8]); Dive phase (from CF_IC_ to Ball [2,7,8]); CF stance phase (from CF_IC_ to CF_TO_ [8]); IF stance phase (from IF_IC_ to IF_TO_ [8]) (Figure 4). 

For each sub-phase, the time elapsed was computed. The kinematics of the centre of mass (CoM) were extracted and analysed in terms of peak velocity magnitude and linear acceleration range in the entire movement and in each sub-phase separately for PS and nPS [2,5]. Joint kinematics of ipsilateral and contralateral (lower and upper) limbs were separately investigated in each sub-phase.

### 2.4. Statistical Analysis

The normal distribution of the data was verified through the Shapiro-Wilk test. The normally distributed continuous data were presented as mean ± standard deviation, whereas the categorical data were presented as a percentage over the total. 

Boxplot was used to inspect the presence of outliers in strength tests data. In the presence of outliers in one of the strength tests, the players were excluded from the kinematical analyses. Strength data were also compared between PS and nPS side through the Student’s *t*-test.

Diving time of the entire movement and each sub-phase was compared according to dive height (high dives and low dives—HLD) and dive side (preferred and non-preferred) through a two-way ANOVA. The η_p_^2^ was reported as a measure of effect size with 0.01, 0.06, 0.14 considered small, moderate, large effects, respectively.

Two separate PCA on high and low dives were conducted to reduce kinematic data dimensionality. The PCA allows the transformation of data waveforms into a small set of features, called principal components (PC), that explain the majority of the variation in the data. The PC scores, the coefficients of the PCs, measure the contribution of a PC to the kinematic waveform shape. Such an approach allows the detection of meaningful differences among different conditions (e.g., in the present study, the diving side preference) when dealing with a limited number of informative variables [10,11,12]. 

An input matrix of 19 participants × 2 sides × 3 trials × 9 joints × 3 axes × 101 (total 310,878 datapoints) was used. The PC scores were extracted for each kinematic variable. If the explained variability of a PC score was >5%, the PC score was compared between PS and nPS side through the student’s *t*-test [10]. Only significant differences (*p* < 0.05) were reported and further investigated [19]. The Cohen’s d effect size was reported alongside *p*-value and was considered small, moderate, and large if 0.2, 0.5, and 0.8, respectively. By examining the shape of waveforms reconstructed through the use of high and low PC scores in the significantly different PCs, it was possible to inspect the characteristics of the differences between diving on PS and nPS without the need to assess all the kinematic variables [10].

The PCA analysis was performed in Matlab (vR2022a), whereas the rest of the statistical analyses were performed in SPSS (v26, IBM, Armonk, NY, USA).

## 3. Results

### 3.1. Time Performance, Strength Performance, CoM Kinematics

Dive time differed between high and low dives in the entire movement and in the sub-phases (*p* < 0.003, η_p_^2^ > 0.041) but did not differ between PS and nPS (*p* > 0.05, η_p_^2^ < 0.007) (Table 2). No outliers were detected in the strength tests and no differences between PS and nPS were noted (Table 3). No differences were found in terms of peak velocity and linear acceleration range between PS and nPS either for the low or the high dive (*p* > 0.05, d < 0.35, Table 4).

### 3.2. Principal Component Analysis

In high dives, 118 features explained at least 5% of overall variability among all the kinematic variables: 49 features belonged to lower limbs (26 ipsilateral, 23 contralateral); 47 features belonged to upper limbs (23 ipsilateral, 24 contralateral); 22 belonged to pelvis, trunk, or head. From this set, 5 PC scores were found to be different between PS and nPS (*p*: 0.047–0.001, d: 0.39–0.66, Table 5). The main differences were found during the take-off phase in hip and knee transverse plane and in trunk and knee frontal plane (Table 5). In particular, players exhibited greater ipsilateral hip internal rotation (Figure 5A and Figure 6A) and trunk ipsilateral tilt on their PS (Appendix A).

In low dives, 104 features explained at least 5% of overall variability among all the kinematic variables: 46 features belonged to lower limbs (25 ipsilateral, 21 contralateral); 39 features belonged to upper limbs (20 ipsilateral, 19 contralateral); 19 belonged to pelvis, trunk, or head. From this set, 6 PC scores were found to be different between PS and nPS (*p*: 0.034–0.003, d: 0.41–0.58, Table 6). The differences regarded the entire movement and were found in hip frontal plane, pelvis and trunk rotations, and shoulder rotation (Table 6). In particular, players exhibited greater ipsilateral hip abduction and contralateral adduction (Figure 5B and Figure 6B), and less pelvis and trunk rotation on their PS (*p* < 0.037, ES 0.40–0.57) compared to the nPS (Appendix A).

## 4. Discussion

The main finding of the present study was that goalkeepers showed kinematic differences in lower limbs, upper limbs, and trunk in frontal and transverse planes between PS and nPS during both high and low dives. Therefore, the first hypothesis was confirmed.

The present study was the first to investigate the full-body kinematics collected on the field in academy goalkeepers. Moreover, it was the first to adopt an innovative approach with data acquired in an ecological environment through wearable sensors and PCA-dimensionality reduction in a complex sport-specific action. Such an approach allowed the description of waveform kinematical differences between the PS and nPS dive, key aspects of goalkeepers’ movement quality [2,3,8].

In high dives, the main differences emerged in the hip-trunk motion: during the PS dive, the ipsilateral limb showed greater hip internal rotation during the initiation phase, followed by a greater trunk tilt during the take-off and ball contact phases with respect to the nPS dive knee (Table 5).

This coordination elicits an efficient strike with the ipsilateral foot, which is not already rotated towards the ball direction (i.e., not limiting the muscle explosiveness) while allowing a faster and larger trunk motion. The contralateral limb was more externally rotated at the hip and in valgus at the knee (Table 5). These rotations likely produced a greater limb push-off to the ground and may indicate that the contralateral limb heavily contributed to the CoM horizontal displacement after the side-step and to the jump efficacy, as previously suggested [8]. All differences occurred always in frontal and transverse planes but never in the sagittal plane, in line with the findings of Spratford et al. [2].

In the low dives to the PS, the hip frontal plane kinematics was compensated between ipsilateral and contralateral limbs, with greater hip abduction in the former and lower in the latter (Table 6). Such an asymmetry elicits the movement towards the ball direction, with the ipsilateral hip as a principal mover. Indeed, in the low dives, the need for an explosive high jump is replaced by a smoother and faster approaching to the ball [2]. Moreover, less ipsilateral rotation for pelvis and trunk was noted. Pelvis and trunk rotations are not desirable when lowering the CoM while moving horizontally [8]. Spratford et al. also found greater pelvis and trunk rotation in nPS diving and correlated it with greater knee joint moments and lower CoM velocity [2]. Such differences suggest more efficient kinematic patterns and inter-joint coordination when diving on the PS compared to the nPS. 

Currently, there are no guidelines regarding goalkeeper’s kinematics describing the dive phases which may help coaches to improve goalkeepers’ dive training and performance. One study focused on the preparatory phase (initial posture): it was stated that a stance width of 75% of the total leg length leads to a better dive performance (timewise) [8]. It was also suggested that a proximal-to-distal sequence (hip-knee-ankle) allows the athlete to generate a higher performance in a vertical jump [20]. However, the vertical jump is not comparable with goalkeepers’ diving saves. Moreover, Ibrahim et al. found that proximal-to-distal sequence during the save dives was present in ipsilateral leg but not in the contralateral [3]. Only one study compared the kinematical pattern of PS and nPS dive in a laboratory setting: in accordance with the findings of the present study, the authors underlined the importance of transverse plane kinematics (mostly pelvis and trunk) to discriminate between a PS and nPS dive [2]. A dimensionality reduction approach based on real world data might be useful to goalkeepers’ coaches to define the dominant differences in diving motion between PS and nPS and improve training effectiveness in academies [2,5].

Notably, the present study did not identify outliers in strength performance among the players nor differences between PS and nPS (Table 3). These results suggest the absence of intra-subject and inter-subject muscular/sprint deficits, thus, the homogeneity of the present study cohort in terms of physical readiness, as assessed in some of the most frequently used strength tests [1]. Moreover, time-performance differences were noted only between high and low dives but not between PS and nPS (Table 2). The difference between high and low dive is in line with the current literature [2]. This aspect suggests that, due to the intrinsic differences in time, muscle activation, and movement technique required, high and low dive should be always investigated as separate entities. The lack of time differences between PS and nPS suggest an overall adequate performance of the players in this game-resembling setup. Similarly, no effect of diving side was noted on CoM velocity and acceleration (Table 4), in line with Spratford et al. [2]. The CoM velocity and acceleration are key indicators for dives performance [2,8,21]: high-level goalkeepers have faster reaction time and could propel faster in the direction of the ball than lower-level goalkeepers [1,2,21]. In the first data-driven intervention study assessing goalkeepers’ biomechanics, Ibrahim et al. recently demonstrated an improvement of CoM velocity after a 12 weeks’ dedicated training program in elite players [22]. However, it is complicated to assess the performance of the goalkeepers only by considering CoM velocity and acceleration and it is, most importantly, hard to transfer it into a dedicated training. 

Overall, the lack of differences in strength, time, and CoM performance allowed the reduction of the risk of bias for the movement analysis and, therefore, an inspection of the differences in joint kinematics between PS and nPS as purely related to the players’ diving movement technique. The use of PCA enhanced the description of the biomechanical characteristics of a diving side and the identification of such underlying relevant differences. Moreover, differences were found in both limbs, in line with the second hypothesis, although mainly in contralateral (lower or upper) ones. Previous literature did not focus on the different contribution of ipsilateral and contralateral limbs to the diving side. The present study might suggest that the ipsilateral limb contributes more to the movement-directional control while the contralateral limb to the jump propulsion. This might also explain why high and low dives could not be considered similar in terms of timing and differences between PS and nPS. Such a different contribution could be object of dedicated training to optimize both the control and propulsion of the diving. The use of standard data analysis based on amplitude-punctual comparisons might not allow the identification of such contributions. Instead, PCA allows the investigation of shape differences containing most of the data variability, without being limited to a punctual analysis [9,10]. In a game situation, the full-body movement technique could make the difference between saving and conceding a goal. An effective training should embrace the multifactorial aspects of the diving save gesture but stay focused on the relevant differences between PS and nPS. This is particularly true in football academies, where movement techniques are trained more extensively. 

From a technical point of view, the use of PCA might help to reduce the setup complexity (i.e., the number of sensors) and at the same time save the most informative differences among players. Future studies assessing the concurrent validity of a limited number of sensors against a full-body setup might provide valuable information and practical implications on this topic. Through a simplified setup, the adoption of quantitative assessment in an ecological environment might be broader and more user-friendly, as for other outdoor applications [13,14,15,16]. The present study results might suggest that a limited number of sensors aiming to capture only hip, pelvis and trunk kinematics, specifically in the first part of the dive, might be necessary to explain most of the differences between PS and nPS dive techniques.

The present study has some limitations. First, the sample size did not allow side inter-age or inter-experience players’ comparison. Moreover, the sample was composed of only male players. However, the present study cohort was larger, mostly double, of the current literature on similar topics and was not biased by performance discrepancies between the players. Second, in line with previous literature, the analysis was conducted in a single session, and it was not possible to assess the day-to-day reliability of the kinematical measures. Moreover, dives were only simulated (no ball kick) and were proposed in a non-randomized order. Hence, although the tests were conducted on a real football pitch, the unpredictability of the game situation could only be partially reproduced. Third, no muscle data were collected. The analysis of muscle synergies would have provided a better understanding of the differences between PS and nPS dives and should be the object of future investigations. However, the use of standard surface EMG units would have been problematic on the field, and should be replaced by more ecological devices, e.g., EMG shorts. The sample frequency (60 Hz) might be suboptimal for these types of movements. This was required by the manufacturer to obtain a full-body kinematical acquisition and allow the entire movement to be inspected. Future studies might consider the use of a higher sample frequency to focus on the biomechanics of the players’ reaction. To make the data easier and simpler, the present study did not take into account segment-angular velocities and accelerations. These aspects have been recently related with joint load and might add value to the description of goalkeepers’ movement technique. This might be the object of future investigations [13,23]. Finally, the use of PCA has been currently limited to gait and jump tasks [9,10,24]. Thus, although theoretically correct, the results derived by such an approach should be interpreted with caution. To the best of the authors’ knowledge, this is the first study adopting PCA with wearables to assess a complex sport-specific task.

## 5. Conclusions

Goalkeepers showed kinematic differences in lower limbs, upper limbs, and trunk in frontal and transverse planes between preferred and non-preferred side during both high and low dives. An innovative approach with wearable sensors plus PCA-dimensionality reduction was proposed. The analysis of relevant quantitative characteristics of goalkeepers’ movement technique might improve training effectiveness.

## Figures and Tables

**Figure 1 sensors-22-07519-f001:**
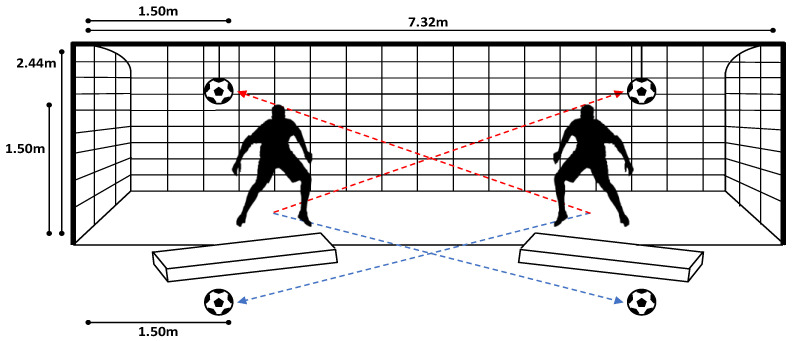
Set-up adopted for the data collection. The football goal was in a regular football pitch with natural grass. The mattresses were used to limit the risk of injuries to the goalkeepers and of damage to the wearable units. The balls for high dives were tied to the crossbar through a rope. The goalkeepers’ starting position was backwards the goal line (as showed by the shape) to allow a more natural diving moment (especially for high dives). The red lines indicate the direction of the high dives, and the blue lines indicate the low dives.

**Figure 2 sensors-22-07519-f002:**
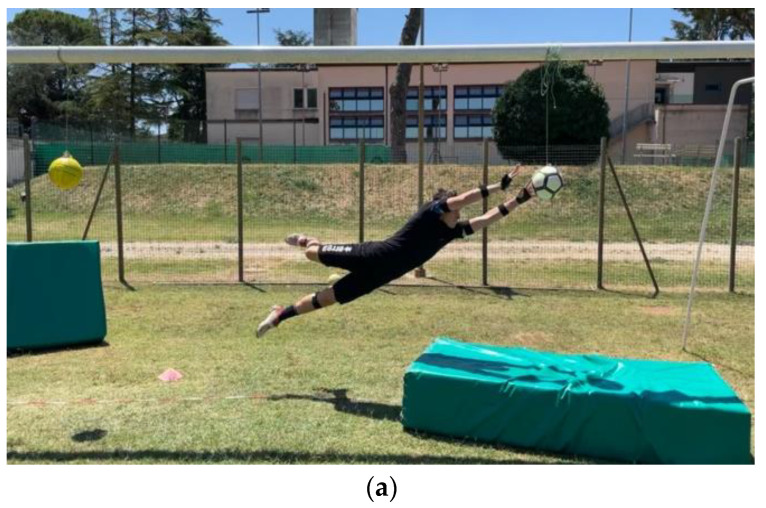
Example of high dive (**a**) and low dive (**b**). The cone was used to standardize the starting position for the save; a rope was used to tie the balls to the crossbar for the high saves; the mattresses were used to avoid injuries to the goalkeepers and damages to the wearable sensor units.

**Figure 3 sensors-22-07519-f003:**
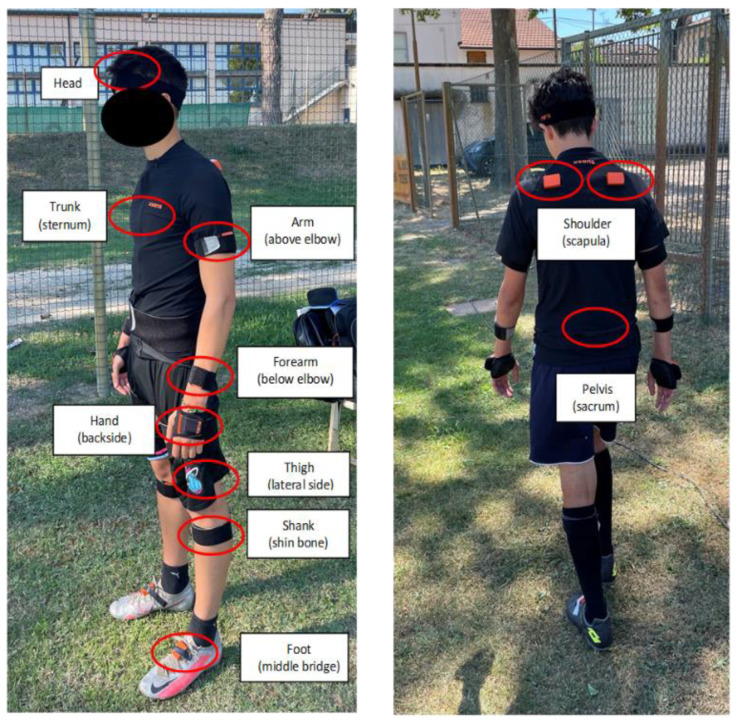
Sensor placement—bilaterally on feet (middle bridge); shanks (shin bone); thighs (lateral side); shoulders (scapulae); arms (lateral above elbow); forearms (lateral below elbow); hands (backside); pelvis (sacrum); trunk (sternum); head.

**Figure 4 sensors-22-07519-f004:**
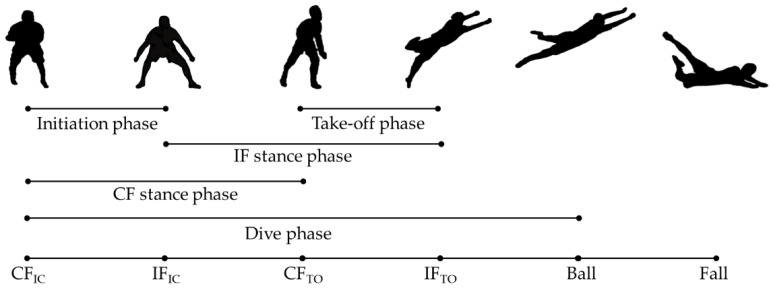
Dive phase division. Note: CF_IC_: Contralateral foot initial contact; IF_IC_: Ipsilateral foot initial contact; CF_TO_: Contralateral foot toe-off; IF_TO_: Ipsilateral foot toe-off; Ball: the contact with the ball; Fall: contact with the ground (mattress).

**Figure 5 sensors-22-07519-f005:**
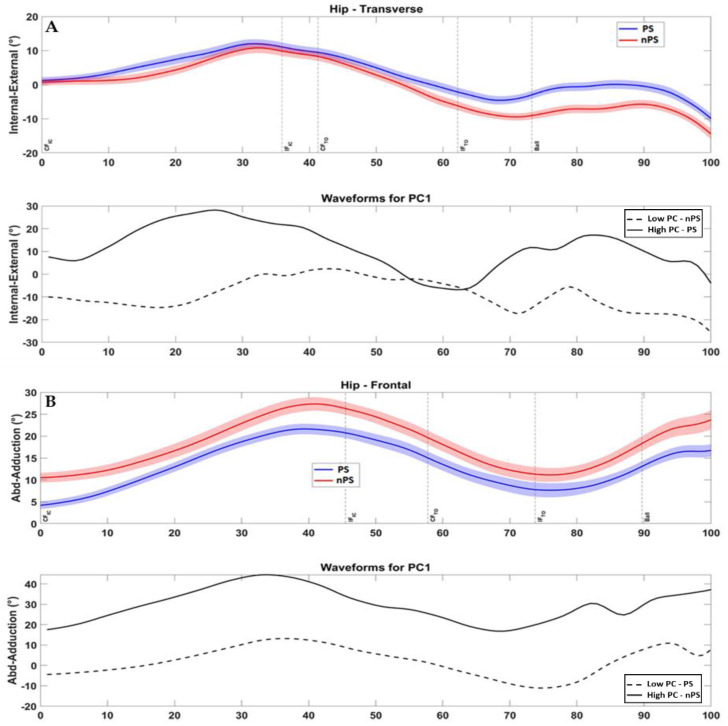
(**A**) High dive—Lower Limb, Ipsilateral—in the bottom row, preferred side waveform is represented by the high PC (solid line) for this kinematic variable; (**B**) Low dive—Lower Limb, Contralateral: in the bottom row, preferred side waveform is represented by the low PC (dashed line) for this kinematic variable. Note: PS = preferred diving side; nPS = non-preferred diving side; PC = principal component; IC = initial contact; TO = toe-off, CF = contralateral foot; IF = ipsilateral foot; Ball = ball contact.

**Figure 6 sensors-22-07519-f006:**
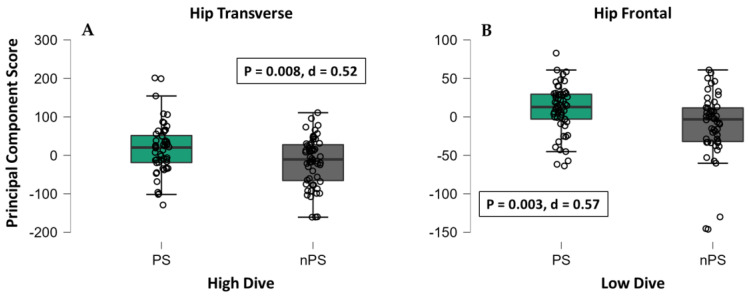
Example Principal Component scores for preferred and non-preferred sides in high dive (**A**) and low dive (**B**). Note: PS = preferred diving side; nPS = non-preferred diving side; the full list of *p*-values and Cohen’s d is reported in Table 5 and Table 6.

**Table 1 sensors-22-07519-t001:** Demographic data, mean ± SD [range].

**Number of Players**	19
**Age (years)**	16.5 ± 3.0 [13–21]
**Height (cm)**	176.0 ± 7.5 [158.6–187.1]
**Mass (kg)**	70.2 ± 8.9 [50.8–86.1]
**BMI**	22.6 ± 2.2 [19.5–27.2]
**Preferred Side ^1^ (r/L)**	9/10

^1^ Note: Preferred diving side indicated by each player.

**Table 2 sensors-22-07519-t002:** Time performance in high and low dives according to the preferred diving side.

Descriptives, Mean (SD)						Statistical Analysis (2-Way ANOVA)									
Phase (s)	Overall	High Dive		Low Dive		Global				HLD		Side		HDL × Side	
		PS	nPS	PS	nPS	R^2^	R^2^ adj.	*p*	η_p_^2^	*p*	η_p_^2^	*p*	η_p_^2^	*p*	η_p_^2^
Initiation	0.49 ± 0.19	0.45 ± 0.16	0.45 ± 0.16	0.51 ± 0.22	0.55 ± 0.18	0.05	0.03	0.020	0.05	0.003	0.04	n.s.	0.00	n.s.	0.00
Take-off	0.22 ± 0.09	0.26 ± 0.07	0.25 ± 0.06	0.18 ± 0.09	0.19 ± 0.09	0.17	0.16	0.000	0.17	0.000	0.17	n.s.	0.00	n.s.	0.00
CF stance	0.59 ± 0.20	0.51 ± 0.17	0.52 ± 0.17	0.64 ± 0.21	0.68 ± 0.17	0.14	0.12	0.000	0.14	0.000	0.13	n.s.	0.01	n.s.	0.00
IF stance	0.31 ± 0.08	0.32 ± 0.07	0.31 ± 0.06	0.31 ± 0.10	0.31 ± 0.08	0.00	−0.01	n.s.	0.00	n.s.	0.00	n.s.	0.00	n.s.	0.00
Dive	0.96 ± 0.23	0.90 ± 0.23	0.91 ± 0.23	0.99 ± 0.24	1.03 ± 0.21	0.05	0.04	0.009	0.05	0.001	0.05	n.s.	0.00	n.s.	0.00
Total	1.19 ± 0.23	1.24 ± 0.23	1.25 ± 0.22	1.12 ± 0.26	1.17 ± 0.21	0.05	0.04	0.009	0.05	0.002	0.04	n.s.	0.01	n.s.	0.00

Note: HLD = high dives and low dives; PS = preferred side, nPS = non-preferred side; n.s. = not significant.

**Table 3 sensors-22-07519-t003:** Strength tests.

**Maximal Double-Leg Hop (cm)**	208.3 ± 22.9
**Maximal single-leg hop, PS (cm)**	180.2 ± 22.4
**Maximal single-leg hop, nPS (cm)**	179.6 ± 21.8
**Maximal 5-m frontal sprint (s)**	1.3 ± 0.1
**Maximal 5-m lateral sprint, PS (s)**	1.7 ± 0.1
**Maximal 5-m lateral sprint, nPS (s)**	1.7 ± 0.1

Note: the results are reported as mean ± standard deviation. PS = preferred side, nPS = non-preferred side; n.s. = not significant; no differences were found between the PS and nPS side tests.

**Table 4 sensors-22-07519-t004:** Velocity and acceleration of the CoM during high and low dives.

	Center of Mass in High Dive				Center of Mass in Low Dive			
	PS	nPS	*p*-Value	d	PS	nPS	*p*-Value	d
**Velocity (m/s, peak)**								
Initiation	2.8 ± 0.2	4.1 ± 5.6	n.s.	0.32	3.0 ± 0.9	3.4 ± 4.0	n.s.	0.15
Take-off	3.5 ± 0.3	3.8 ± 1.7	n.s.	0.29	3.7 ± 0.4	3.9 ± 1.6	n.s.	0.15
CF stance	2.9 ± 0.3	4.3 ± 5.6	n.s.	0.32	3.3 ± 0.9	3.7 ± 4.0	n.s.	0.14
IF stance	3.5 ± 0.3	3.9 ± 1.8	n.s.	0.31	3.7 ± 0.4	4.0 ± 2.6	n.s.	0.18
Dive	3.5 ± 0.3	4.9 ± 5.4	n.s.	0.35	4.1 ± 0.8	4.6 ± 3.9	n.s.	0.17
Total	4.2 ± 0.3	5.6 ± 5.3	n.s.	0.35	4.2 ± 0.8	4.6 ± 3.9	n.s.	0.17
**Acceleration (m/s^2^, range)**								
Initiation	12.6 ± 2.9	12.1 ± 3.0	n.s.	0.15	11.0 ± 3.2	11.3 ± 3.4	n.s.	0.09
Take-off	9.5 ± 4.2	9.0 ± 3.8	n.s.	0.13	6.5 ± 2.9	7.6 ± 4.4	n.s.	0.30
CF stance	13.8 ± 3.1	12.8 ± 3.2	n.s.	0.32	12.5 ± 2.6	12.3 ± 3.4	n.s.	0.07
IF stance	12.5 ± 4.5	11.8 ± 4.0	n.s.	0.16	10.0 ± 3.0	10.6 ± 4.5	n.s.	0.17
Dive	17.6 ± 3.7	17.1 ± 3.6	n.s.	0.15	18.7 ± 8.2	22.5 ± 16.1	n.s.	0.30
Total	20.7 ± 8.9	22.4 ± 13.1	n.s.	0.15	46.8 ± 17.2	46.6 ± 20.4	n.s.	0.02

Note: PS = preferred side, nPS = non-preferred side; n.s. = not significant.

**Table 5 sensors-22-07519-t005:** Principal Components differences in High Dive kinematics.

Kinematic Variable	Explained Variability (%)	PC Score PS	PC Scoren PS	*p*-Value	Effect Size	Feature Explanation
**Lower Limb, Ipsilateral**						
Hip intra-extra rotation	49.3	17.7 ± 67	−16.4 ± 63.9	0.008	0.52	Greater internal rotation peaks
**Lower Limb, Contralateral**						
Hip intra-extra rotation	43.6	5.8 ± 23.5	−5.4 ± 28.4	0.029	0.43	Greater external rotation and ROM before take-off
Knee varus-valgus	26.2	6.2 ± 23.4	−5.7 ± 15.8	0.002	0.60	More valgus before take-off
Knee intra-extra rotation	62.3	−27.7 ± 83.7	25.7 ± 78.6	0.001	0.66	Less ROM and greater external rotation before take-off
**Pelvis/Trunk/Head**						
Trunk ipsilateral tilt	11.1	−6.4 ± 28.5	5.9 ± 34.7	0.047	0.39	Greater ipsilateral tilt during take-off and ball contact

Note: PS = preferred side, nPS = non-preferred side; n.s. = not significant.

**Table 6 sensors-22-07519-t006:** Principal Components differences in Low Dive kinematics.

Kinematic Variable	Explained Variability (%)	PC Score PS	PC Score nPS	*p*-Value	Effect Size	Feature Explanation
**Lower Limb, Ipsilateral**						
Hip abd-adduction	17.2	10.2 ± 31.0	−11.2 ± 43.3	0.003	0.57	Greater abduction with less ROM
**Lower Limb, Contralateral**						
Hip abd-adduction	58.6	−22.3 ± 71.3	24.4 ± 90.1	0.003	0.58	Lower abduction (all movement)
**Upper Limb, Contralateral**						
Shoulder intra-extra rotation	21.3	−23.6 ± 96.4	25.8 ± 141.1	0.034	0.41	Greater internal rotation at take-off and ball contact
Wrist flexion-extension	11.9	10.6 ± 45.7	−11.6 ± 53.8	0.022	0.45	Less extension during initiation and take-off
**Pelvis/Trunk/Head**						
Pelvis ipsilateral rotation	14.1	−3.6 ± 14.3	4.0 ± 18.4	0.018	0.46	Less ipsilateral rotation (all movement)
Trunk ipsilateral rotation	14.0	−8.6 ± 30.3	9.4 ± 42.9	0.012	0.49	Less ipsilateral rotation (all movement)

Note: PS = preferred side, nPS = non-preferred side; n.s. = not significant.

## Data Availability

Not applicable.

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
