# Peer review of "On-Field Biomechanical Assessment of High and Low Dive in Competitive 16-Year-Old Goalkeepers through Wearable Sensors and Principal Component Analysis"

_sensors, 2022, doi:10.3390/s22197519_

Round 1
Reviewer 1 Report
Mihnea-Ion MARIN
E-mail: [email protected], [email protected]
Thank you for the opportunity to review this interesting article.
Peer-review report 1875037
The paper On-Field Biomechanical Assessment of High and Low Dive in Competitive Goalkeepers through Wearable Sensors and Principal Component Analysis, written by authors Stefano Di Paolo, Francesco Santillozzi, Raffaele Zinno, Giuseppe Barone, and Laura Bragonzoni, presents biomechanical investigations of dive techniques of competitive goalkeepers, using Xsens wearable sensors.
Below are a few remarks, the clarification of which, would allow a better understanding of the text and complete this interesting study:
1.In the Introduction: a detailed review of the studies that have dealt with the various assessments of football goalkeepers has been made, concluding that to date no measurements of these subjects with wearable sensors have been made; nothing has been mentioned about the assessment with this type of sensor of football players in other posts. I think it would be useful to review the findings of such studies.
2.For clarification, please state the following:
-in the Data collection (lines 104-105), it was specified taking a video capture and synchronize it with the data collected from the sensors, without specifying the smartphone synchronization software. As far as I know, the Awinda box allows synchronization using a specialized device, not directly via a smartphone. What type of specialized software was used to synchronize the video image with the signals collected by the sensors?
-how was achieved the alignment of the local reference system of each sensor with the global reference system of the football field?
-what are the characteristic points where the Xsens sensors were placed on the human body?
-why were the stress tests necessary: to detect outliers (lines 179-183), or to demonstrate the homogeneity of the group of evaluated subjects (lines 262-263)?
-why did the study only analyse linear acceleration in the two planes and not data on the angular accelerations of the various characteristic points where the sensors were placed?
- in the end of the Introduction, two hypotheses were clearly stated, but in the Discussion chapter the confirmation of the first hypothesis (line 218) was clearly presented, but the conclusion related to the second hypothesis was too briefly treated.
The discussions are detailed and coherent.
The final conclusions are coherent and summarize the fact that monitoring the movements of the Competitive Goalkeepers through the use of sensors is possible through the use of wearable sensors thanks to recent advances in technology.
The figures are clear and easy to understand.
The paper uses grammatically and academically appropriate language, of substance, and is easy to understand. The paper is relevant for specialists because presents a new approach to monitoring goalkeepers.
The article is an interesting study, which can be published after incorporating the reviewers' remarks.
Author Response
Dear Reviewer,
Thanks for the opportunity to revise our manuscript in light of such precious and focused comments. Some comments helped refine the paper’s workflow, others helped extend the discussion section with profitable practical implications for the readers. We answered point-by-point in this cover letter and modified the manuscript accordingly. All the changes were reported in this cover letter to facilitate the revision process and meanwhile tracked in the manuscript through the track changes (new sentence highlighted). Extensive reasons for the comments’ rebuttal were given where necessary. We feel that the manuscript has now improved in clarity, completeness, and overall quality, and hope it now reaches the standard for publication in Sensors.
Reviewer 1
Thank you for the opportunity to review this interesting article.
Peer-review report 1875037
The paper On-Field Biomechanical Assessment of High and Low Dive in Competitive Goalkeepers through Wearable Sensors and Principal Component Analysis, written by authors Stefano Di Paolo, Francesco Santillozzi, Raffaele Zinno, Giuseppe Barone, and Laura Bragonzoni, presents biomechanical investigations of dive techniques of competitive goalkeepers, using Xsens wearable sensors.
Below are a few remarks, the clarification of which, would allow a better understanding of the text and complete this interesting study:
- In the Introduction:
a detailed review of the studies that have dealt with the various assessments of football goalkeepers has been made, concluding that to date no measurements of these subjects with wearable sensors have been made; nothing has been mentioned about the assessment with this type of sensor of football players in other posts. I think it would be useful to review the findings of such studies.
A: Thanks for the comment. We agree with the Reviewer’s comment and extended the introduction with 4 studies regarding the biomechanical assessment of outfield players on the field through wearable sensors. We added as follows: “Previous studies investigated on-field biomechanics of outfield football players during sport-specific activities, such as agility tasks and side-games, and put movement characteristics (also extracted through PCA dimensionality reduction) in relation to players’ coordinative abilities, training load, fatigue, and lower-limb injury risk [13–16].”
- For clarification, please state the following:
-in the Data collection (lines 104-105), it was specified taking a video capture and synchronize it with the data collected from the sensors, without specifying the smartphone synchronization software. As far as I know, the Awinda box allows synchronization using a specialized device, not directly via a smartphone. What type of specialized software was used to synchronize the video image with the signals collected by the sensors?
A: Thanks for the comment. We notice from this and another Reviewer’s comment that our explanation was somehow misleading. We did not use a trigger between the Xsens Awinda and the smartphone (an iPhone 12, Apple Inc., Los Altos, CA, US – added in the manuscript). The exact foot contact phases were detected through the “foot contact point detection” variable provided by the Xsens software and trough visual inspection of the avatar reconstruction. A frontal view smartphone video was recorded for each trial and was used alongside the avatar movement reconstruction in the Xsens software environment to inspect the movement and support the detection of the different sub-phases and dive’s peculiarities (undesired initial steps, hand-ball contact, etc.). We rephrased as follows: “The frames were detected through the “foot contact point detection” variable and trough visual inspection of the avatar reconstruction provided in the manufacturer software environment. The smartphone video captures were used to support the detection of the foot contact phases and dive peculiarities (undesired initial steps, hand-ball contact, etc.).”
-how was achieved the alignment of the local reference system of each sensor with the global reference system of the football field?
A: Thanks for the question. The alignment of local sensors’ reference systems with the global reference system is usually computed internally in the Xsens software by a fusion engine process. The manufacturer describes the process as follows: “The MVN Fusion Engine calculates the position and orientation, and other kinematic data of each body segment, with respect to an earth-fixed reference co-ordinate system. By default, the earth-fixed reference co-ordinate system used is defined as a right-handed Cartesian co-ordinate system. The axes of the Global reference system are defined as: X (red) pointing to the local magnetic North; Y (green) according to right-handed coordinate system (West); Z (blue) pointing up. The axes of each body frame are aligned with this Global reference frame during the T-Pose (calibration phase)”, (https://www.xsens.com/hubfs/Downloads/usermanual/MVN_User_Manual.pdf).
-what are the characteristic points where the Xsens sensors were placed on the human body?
A: Thanks for the question. We reported the characteristics points where the sensors are placed in the Data collection section as follows: “In brief, sensors were placed bilaterally on feet (middle bridge), shanks (shin bone), tights (lateral side), shoulders (scapulae), arms (lateral above elbow), forearms (lateral below elbow), and hands (backside); the pelvis sensor was placed on the sacrum, the trunk sensor was placed on the sternum, and the head sensor was placed through a headband”. Furthermore, also according to another Reviewer’s request, we added a figure reporting the example of a subject wearing all the sensors with relative description.
-why were the stress tests necessary: to detect outliers (lines 179-183), or to demonstrate the homogeneity of the group of evaluated subjects (lines 262-263)?
A: Thanks for the comment. The strength tests data collected in this study are common field tests to assess the physical status of the football goalkeepers. The aim for reporting such info was to limit as much as possible the confounding factors that could have influenced the kinematical analysis. Indeed, we looked for 1. outliers in the data, thus if possible subjects with too high or too low performances (and relate muscular/sprint deficits) were present; 2. Differences between PS and nPS in strength tests. Overall, the presence of no outliers nor side-to-side differences confirmed the homogeneity of our cohort towards strength and sprinting capacities and corroborated that the kinematical differences that we found were purely related to the diving movement technique. We agree with the Reviewer that these explanations were not detailed enough in the manuscript, thus we modified as follows: “These results suggest the absence of intra-subject and inter-subject muscular/sprint deficits, thus, the homogeneity of the present study cohort in terms of physical readiness, as assessed through some of the most frequently used strength tests”.
-why did the study only analyse linear acceleration in the two planes and not data on the angular accelerations of the various characteristic points where the sensors were placed?
A: Thanks for the comment. The linear acceleration was assessed for the Center of Mass (CoM) only. This data felt into an analysis of “Center of Mass performance”, alongside CoM peak velocity and CoM time. We used these measures to be in line with the current literature that focused mostly on the biomechanics of the CoM, instead of the angular kinematics. We agree that segmental angular accelerations might be thrilling information to investigate goalkeeper dives, and that might be the object of investigation in future research. For the present paper, we feel this addition would add complexity to the readers’ interpretability of the manuscript and decided to keep the angular kinematics only in the PCA investigation. We enlarged our discussion section to describe this concept as “To facilitate results interpretability and avoid data overload, the present study did not take into account segment angular velocities and accelerations. These aspects have been recently related to joint load and might add value to the description of goalkeepers’ movement technique, thus might be the object of future investigations [13,23]”.
- in the end of the Introduction, two hypotheses were clearly stated, but in the Discussion chapter the confirmation of the first hypothesis (line 218) was clearly presented, but the conclusion related to the second hypothesis was too briefly treated.
A: Thanks for the comment. The first hypothesis was the one with the strongest rationale and literature background, therefore we spent more time on it in the discussion section. However, we agree with the Reviewer on the need to further discuss the second hypothesis and extended the section with further comments on the practical implications of this finding as follows: “… This might also explain why high and low dives could not be considered similar in terms of timing and differences between PS and nPS. Such a different contribution could be the object of dedicated training to optimize both control and propulsion of the diving.”
The discussions are detailed and coherent.
The final conclusions are coherent and summarize the fact that monitoring the movements of the Competitive Goalkeepers through the use of sensors is possible through the use of wearable sensors thanks to recent advances in technology.
The figures are clear and easy to understand.
The paper uses grammatically and academically appropriate language, of substance, and is easy to understand. The paper is relevant for specialists because presents a new approach to monitoring goalkeepers.
The article is an interesting study, which can be published after incorporating the reviewers' remarks.
A: Thanks for the appreciation and for all the valuable comments and suggestions provided.

Reviewer 2 Report
This paper investigates the characteristics of goalkeepers’ dive in PS and nPS side. PCA is conducted to reduce data dimensionality. The authors claim that biomechanical investigation of diving techniques improve the performance of training. The idea of the paper is interesting; however, I have the following comments/suggestions to further improve the quality of the paper.
1. There are some minor grammatical errors and typos. The paper needs to be revised in terms of language.
2. The related work needs some improvement. Some relevant works based on wearable sensors need to be presented.
3. Figure 2 is needed to refine to clearly depict the experimental data collection process.
4. It is necessary to describe the configuration of the wearable sensors and to add a figure of the subject wearing the sensors in the corresponding positions.
5. Can the author justify the use of these many sensors in their proposed solution? Can we detect the user's posture using a smaller number of sensors? There are some related works to study posture characteristics with fewer sensors.
6. I also have concerns about the section of data processing. The authors have not provided many details, such as how the sensor and video data are synchronized and how the sub-phases are divided. In addition, please add a figure describing the dive phase division.
Author Response
Dear Reviewer,
Thanks for the opportunity to revise our manuscript in light of such precious and focused comments. Some comments helped refine the paper’s workflow, others helped extend the discussion section with profitable practical implications for the readers. We answered point-by-point in this cover letter and modified the manuscript accordingly. All the changes were reported in this cover letter to facilitate the revision process and meanwhile tracked in the manuscript through the track changes (new sentence highlighted). Extensive reasons for the comments’ rebuttal were given where necessary. We feel that the manuscript has now improved in clarity, completeness, and overall quality, and hope it now reaches the standard for publication in Sensors.
Reviewer 2
This paper investigates the characteristics of goalkeepers’ dive in PS and nPS side. PCA is conducted to reduce data dimensionality. The authors claim that biomechanical investigation of diving techniques improve the performance of training. The idea of the paper is interesting; however, I have the following comments/suggestions to further improve the quality of the paper.
- There are some minor grammatical errors and typos. The paper needs to be revised in terms of language.
A: Thanks for the comment. The manuscript has now been revised by a native English speaker for grammatical errors and typos.
- The related work needs some improvement. Some relevant works based on wearable sensors need to be presented.
A: Thanks for the comment. According to this and the other Reviewer’s comments, we enlarged both the introduction and discussion section including more references to wearable sensors papers in football on-field testing. The following references were added:
- Bastiaansen, B.J.C.; Vegter, R.J.K.; Wilmes, E.; de Ruiter, C.J.; Lemmink, K.A.P.M.; Brink, M.S. Biomechanical Load Quan-tification Using a Lower Extremity Inertial Sensor Setup During Football Specific Activities. Sports Biomech. 2022, 1–16, doi:10.1080/14763141.2022.2051596.
- Di Paolo, S.; Nijmeijer, E.; Bragonzoni, L.; Dingshoff, E.; Gokeler, A.; Benjaminse, A. Comparing Lab and Field Agility Kinematics in Young Talented Female Football Players: Implications for ACL Injury Prevention. J. Sport Sci. 2022, 1–10, doi:10.1080/17461391.2022.2064771.
- Di Paolo, S.; Zaffagnini, S.; Pizza, N.; Grassi, A.; Bragonzoni, L. Poor Motor Coordination Elicits Altered Lower Limb Bio-mechanics in Young Football (Soccer) Players: Implications for Injury Prevention through Wearable Sensors. Sensors 2021, 21, 4371, doi:10.3390/s21134371.
- Wilmes, E.; de Ruiter, C.J.; Bastiaansen, B.J.C.; Zon, J.F.J.A. van; Vegter, R.J.K.; Brink, M.S.; Goedhart, E.A.; Lemmink, K.A.P.M.; Savelsbergh, G.J.P. Inertial Sensor-Based Motion Tracking in Football with Movement Intensity Quantification. Sensors 2020, 20, 2527, doi:10.3390/s20092527.
- Pratt, K.A.; Sigward, S.M. Inertial Sensor Angular Velocities Reflect Dynamic Knee Loading during Single Limb Loading in Individuals Following Anterior Cruciate Ligament Reconstruction. Sensors 2018, 18, E3460, doi:10.3390/s18103460.
If this Reviewer believes that other relevant works need to be reported, we’ll welcome the suggestion.
- Figure 2 is needed to refine to clearly depict the experimental data collection process.
A: Thanks for the suggestion. We refined the Figure 2 by adding as much details as possible to describe the data collection process without the need to refer to the main text.
- It is necessary to describe the configuration of the wearable sensors and to add a figure of the subject wearing the sensors in the corresponding positions.
A: Thanks for the comment. According to this and another Reviewer’s requests, we enlarged the description of the wearables’ configuration and placement in the Data collection section as follows: “In brief, sensors were placed bilaterally on feet (middle bridge), shanks (shin bone), tights (lateral side), shoulders (scapulae), arms (lateral above elbow), forearms (lateral below elbow), and hands (backside); the pelvis sensor was placed on the sacrum, the trunk sensor was placed on the sternum, and the head sensor was placed through a headband”. Furthermore, we added a figure reporting the example of a subject wearing all the sensors with a relative description.
- Can the author justify the use of these many sensors in their proposed solution? Can we detect the user's posture using a smaller number of sensors? There are some related works to study posture characteristics with fewer sensors.
A: Thanks for the comment. We strongly agree with the possibility to reduce the number of sensors for sports applications. The adoption of a simplified sensors setup to detect posture characteristics without losing accuracy is a hot topic in the on-field-sports biomechanics community. We believe this point will foster the use of quantitative data in the ecological environment. Our study somehow fits into this aim and direction, since we used a dimensionality reduction approach (PCA) to keep only informative differences from the full-body kinematics. From our results, an indication might be the need for a limited number of sensors to capture only hip, pelvis and trunk kinematics, specifically in the first part of the dive. However, the present study was the first to investigate full-body kinematics with sensors on the field during such a complex, sport-specific, and multidirectional task. Thus, we believed it was necessary to obtain data from all the joints possible and depict as much information as possible, as a first guess. For example, we did not know (neither from in-lab studies) the influence of upper limbs on the different diving side biomechanics. We strongly believe that future studies assessing the concurrent validity of a limited number of sensors against a full-body setup might provide valuable information and practical implications for this topic. We enlarged the discussion section to cover all these points as follows: “From a technical point of view, the use of PCA might help reduction the setup complexity (i.e., the number of sensors) while keeping the most informative differences among players. Future studies assessing the concurrent validity of a limited number of sensors against a full-body setup might provide valuable information and practical implications on this topic. Through a simplified setup, the adoption of quantitative assessment in ecological environment might be broader and more user-friendly, as for other outdoor applications [13–16]. The present study results might suggest that a limited number of sensors aiming to capture only hip, pelvis and trunk kinematics, specifically in the first part of the dive, might be necessary to explain most of the differences between PS and nPS dive technique.”
- I also have concerns about the section of data processing. The authors have not provided many details, such as how the sensor and video data are synchronized and how the sub-phases are divided. In addition, please add a figure describing the dive phase division.
A: Thanks for the comment. During the revision process, according to this and another Reviewers’ valuable suggestions, we enlarged the papers’ methodological description, especially according to the use of video, the goal setup, the wearable placements, and the variables under investigation. According to the Reviewer’s suggestion, we provided a detailed image of the dive phase division in Figure 3.

Round 2
Reviewer 2 Report
This is an significant study on fall detection of older adults based on wearable sensors systems. The paper is well organized with clear figures and supporting charts. It is well written and the content is comprehensive. Overall, I agree to publish this work in Sensors.
Author Response
Thanks for the comment.
However, this paper is about goalkeepers dive performance instead of fall detection in older adults. There might be a mistake in comment delivery.
Thanks